# MicroRNA-Related Polymorphism and Their Association with Fibromyalgia

**DOI:** 10.3390/genes14071312

**Published:** 2023-06-21

**Authors:** Fabian Berg, Dirk A. Moser, Verena Hagena, Fabian Streit, Benjamin Mosch, Robert Kumsta, Stephan Herpertz, Martin Diers

**Affiliations:** 1Department of Genetic Psychology, Faculty of Psychology, Ruhr-University Bochum, Universitätsstraße 150, 44801 Bochum, Germany; fabian.berg@rub.de; 2Department of Psychosomatic Medicine and Psychotherapy, LWL University Hospital, Ruhr University Bochum, 448791 Bochum, Germany; verena.hagena@rub.de (V.H.); martin.diers@rub.de (M.D.); 3Department of Genetic Epidemiology in Psychiatry, Central Institute of Mental Health, Medical Faculty Mannheim, University of Heidelberg, 68159 Mannheim, Germany; fabian.streit@zi-mannheim.de; 4Department of Behavioural and Cognitive Sciences, Laboratory for Stress and Gene-Environment Interplay, University of Luxemburg, Porte des Sciences, L-4366 Esch-sur-Alzette, Luxembourg; robert.kumsta@uni.lu

**Keywords:** fibromyalgia (FM), miRNA, pleiotropy, gene regulation, epigenetics

## Abstract

MicroRNAs are tissue-specific expressed short RNAs that serve post-transcriptional gene regulation. A specific microRNA can bind to mRNAs of different genes and thereby suppress their protein production. In the context of the complex phenotype of fibromyalgia, we used the Axiom miRNA Target Site Genotyping Array to search genome-wide for DNA variations in microRNA genes, their regulatory regions, and in the 3’UTR of protein-coding genes. To identify disease-relevant DNA polymorphisms, a cohort of 176 female fibromyalgia patients was studied in comparison to a cohort of 162 healthy women. The association between 48,329 markers and fibromyalgia was investigated using logistic regression adjusted for population stratification. Results show that 29 markers had *p*-values < 1 × 10^−3^, and the strongest association was observed for rs758459 (*p*-value of 0.0001), located in the *Neurogenin 1* gene which is targeted by hsa-miR-130a-3p. Furthermore, variant rs2295963 is predicted to affect binding of hsa-miR-1-3p. Both microRNAs were previously reported to be differentially expressed in fibromyalgia patients. Despite its limited statistical power, this study reports two microRNA-related polymorphisms which may play a functional role in the pathogenesis of fibromyalgia. For a better understanding of the disease pattern, further functional analyses on the biological significance of microRNAs and microRNA-related polymorphisms are required.

## 1. Introduction

Fibromyalgia (FM) is a chronic pain disorder characterized by widespread musculoskeletal pain and tenderness at specific sites [1]. Patients report symptoms at the joint muscles and tendons on all four quadrants of the body, with increased pain under load. Additional symptoms include general weakness, chronic fatigue, sleep disturbances, cognitive dysfunction, impaired concentration, and reduced mental and physical capacity [2,3]. Patients with FM have reduced pain thresholds and impaired neuronal processing and modulation of pain [4,5,6,7]. The prevalence of fibromyalgia is estimated at around 2–4% in the general population (85–90% are female) and may be caused by a combination of genetic and environmental factors [8,9,10]. Certain environmental factors, such as physical or emotional trauma, may trigger the development of fibromyalgia in individuals who are genetically predisposed to the condition [11,12,13].

MicroRNAs (miRNAs; miR), first described in 1993 [14], are small non-coding single-stranded RNA molecules that play an essential role in gene regulation at the post-transcriptional level. They bind to short segments in the three prime untranslated region (3′-UTR) of mRNA, which leads to degradation or repression of the mRNA and to the inhibition of protein production [15]. MiRNAs are found in large numbers throughout the genome, and to date, more than 2000 different miRNAs have been described, each with tissue-specific expression. These are estimated to regulate one third of all genes, and a single miRNA can modulate the expression of hundreds of different genes. At the same time, a single mRNA or gene can be the target of many different miRNAs so that very large regulatory networks can be assumed in the context of miRNA-mediated gene regulation.

Genetic variation can further affect the regulation of protein expression by microRNAs [16]. Studies have shown that variation in genes that encode microRNAs can alter the regulation of protein expression, leading to abnormal neural development and function, which can contribute to the development of mental disorders [17,18]. Variants in miRNA genes can also have profound effects on miRNA transcription levels and miRNA maturation [19]. For instance, SNPs in miRNA regulatory regions may influence miRNA production. In addition, SNPs in miRNA-coding and miRNA target regions may alter binding affinity of miRNAs by creating or disrupting binding sites [16]. Accordingly, a single miRNA-related SNP potentially alters production or binding of various miRNAs. In addition, genetic variants that alter miRNA binding sites in protein-coding genes have also been linked to mental disorders and drug susceptibility [20,21,22]. Unfortunately, there is no direct evidence of miRNA related genetic variation in the pathology of FM yet. However, the heritability of FM is estimated at around 13.9% in people with European ancestry [23], suggesting involvement of genetic variation. Moreover, first indications of characteristically altered miRNA profiles have been reported in FM patients. Bjersing and colleagues [24] identified nine miRNAs in the cerebrospinal fluid of ten FM patients that showed significantly reduced expression compared to eight healthy control subjects. One of these miRNAs (miR-145-5p) also showed a positive correlation with the cardinal symptoms of pain and fatigue as assessed by the fibromyalgia impact questionnaire (FIQ). Another study showed altered miRNA expression profiles in the blood serum of 20 FM patients compared to 20 healthy controls [25]. Seven miRNAs were significantly less expressed in the FM group, while miRNA miR-320a was increased and slightly negatively correlated with pain according to FIQ. Another group [26] found at least twofold reduced levels in 233 of 1212 analyzed miRNAs in a cohort of eleven FM patients compared to ten control subjects. The expression of five miRNAs was reduced fourfold or more, including miR-145-5p and miR-223-3p, which also showed reduced expression levels [24]. A more recent study, using circulating miRNAs in blood serum and saliva from 14 FM patients and gender- and age-matched controls, demonstrated six significantly downregulated miRNAs, five of which were predictive of the expression of clinical features in patients [27].

In summary, miRNAs, as powerful regulators of gene expression, very likely play a significant role in the pathophysiology of not only FM but also chronic pain disorders [28]. A detailed investigation of the molecular biological processes underlying this regulation therefore not only helps to understand the pathophysiological basis of chronic pain disorders but may even contribute to diagnosing them more reliably and precisely in the future, as well as enabling appropriately effective medication.

The aim of this pilot study was therefore to investigate genome-wide DNA-polymorphisms in miRNA genes and miRNA binding regions on mRNA coding genes in 176 female FM patients and 162 female healthy controls (HC) to identify miRNA-signaling related SNPs associated with FM.

## 2. Materials and Methods

### 2.1. Participants and Procedure

A total of 208 female individuals with FM were recruited mainly through social media support groups. FM diagnoses were obtained by medical professionals and disorders fulfilled the criteria postulated by Wolfe et al. ([29]; see Table 1). Additionally, 202 female HC were recruited via newspaper announcements and face-to-face acquisition at several blood donation events of the German Red Cross. Participants completed a paper and pencil survey that was sent to them by post and collected a mouthwash sample using 10 mL of a non-alcoholic solution. Participation was voluntary and was compensated with EUR 10.

The data collection took place between May 2019 and July 2020. The study was approved by the ethics review board of the Medical Faculty, Ruhr University Bochum (6594-BR). All participants gave written informed consent to participate in the study.

### 2.2. Diagnostic and Clinical Assessment

FM patients completed the West Haven-Yale Multidimensional Pain Inventory (MPI; [30]; German version: [31]), the Chronic Pain Grade Scale (CpG; [32]), the Fibromyalgia Survey Questionnaire (FSQ; [33]), the Fibromyalgia Impact Questionnaire (FIQ-G; German version: [34]), and the Pain-related Self Statements Scale (PRSS; German version: FSS; [35]). FM and HC completed the Center for Epidemiologic Studies Depression Scale (CES-D; [36]; German version: [37]). For a summary of the utilized diagnostic and clinical tools as well as the group values we obtained, see Table 1.

### 2.3. Sample Characteristics

After quality check, 170 patients with FM (aged 50.20 ± 9.45 years, range from 23 to 68 years) and 162 HC (aged 47.46 ± 15.35 years, range from 19 to 81 years) remained in the sample. Age was not significantly different between groups (*t*(261.722) = 1.941, *p* = 0.053), whereas the FM group showed significantly elevated depression symptoms (CES-D: FM: 23.61 ± 7.00, HC: 14.51 ± 4.95, *t*(300.995) = 13.639, *p* < 0.001).

FM patients reported longstanding disease with a mean pain duration of 16.99 (±12.82; range 0.87 to 52.60) years, high scores of pain severity (MPI: 4.07 ± 0.98), and FM impact (FIQ: 56.69 ± 15.37).

### 2.4. DNA Extraction and Quality Controls, and Genotyping Array

DNA was extracted from mouthwash samples using the salting-out method as described by Miller et al. [38]. DNA concentration and purity was photometrically measured (Synergy 2, Biotech/Agilent; Santa Clara, CA, USA) and integrity estimated by agarose gel electrophoresis. A total of 376 samples (197 patients with FM, 179 HC) had the required yield, purity, and integrity, were diluted to a concentration of 15 ng/µL in a volume of 80 µL, and were sent to our partner laboratory Cologne Center for Genomics (CGC). DNA of all subjects was simultaneously analyzed for a total of 237,858 DNA variations distributed in the genome using Axiom miRNA Target Site Genotyping Arrays (Applied Biosystems/Thermo Fisher Scientific; Waltham, MA, USA). This genome-wide array focuses on genotyping SNPs and InDels located in miRNA promoters, miRNA seed sequences, in the 3’UTR of protein-coding genes and in the DNA of miRNA-processing proteins. In a final step, Axiom miRNA Target Site Genotyping Arrays were processed using the GeneTitan^®^ Multi-Channel Instrument, according to the manufacturer’s protocol (Thermo Fischer Scientific, Waltham, MA, USA).

### 2.5. Biostatistical Analysis

The generated array data were first pre-processed using Thermo Fisher Axiom Analysis Suite software. Further quality control was performed using Plink [39]. Here, standard quality control steps were performed using the following exclusion criteria for individuals: >0.02 missingness, heterozygosity rate > |0.20| or sex-mismatch. SNPs were excluded if they deviated from Hardy–Weinberg Equilibrium (HWE) with a *p*-value of <10^−6^, had minor allele frequency <0.01 or missing data >0.02. Relatedness and population stratification were determined using a SNP set filtered for SNP quality (HWE *p* > 0.02, MAF > 0.20, missingness = 0) and LD adjustment (r^2^ = 0.1). In case of cryptic relatedness (pi-hat > 0.20), one subject was randomly removed. To control for population structure, subjects who differed by more than 4.5 standard deviations in the first 20 genetic principal components (PCs) were excluded. As a result, data of 48,329 SNPs were available from 170 fibromyalgia patients and 162 healthy controls.

SNPs related to FM were identified by applying a genome-wide association study (GWAS) approach. In particular, SNPs were tested for association using the logistic regression model incorporated in Plink. Principal components 1–5 from quality control were thereby included as covariates. Bonferroni-corrected genome-wide significance was calculated as *p* < 1.03 × 10^−6^ (0.05/48,329 SNPs). Though according to the strategy implemented by Wilkins and colleagues [40], SNPs with *p*-values < 1 × 10^−3^ were also considered for analysis of miRNA-related functional consequences.

In silico analysis of predicted miRNA-binding alterations due to DNA-polymorphisms were performed by mining the miRNASNP-v3 database [41]. miRNASNP-v3 makes use of TargetScan v7.2 and miRmap v1.1 to determine miRNA target binding sites and to predict SNP-induced gain or loss of miRNA/target pairs [41,42,43]. Accordingly, we examined all of the *p* < 1 × 10^−3^ SNPs for inducing gain or loss of miRNA/target pairs. Subsequently, we checked whether any of the SNP-induced gains or losses of miRNAs were previously related to FM. A total of 39 FM-related miRNAs were identified from manually scanning 26 publications reporting on miRNA expression in FM. The literature search was performed on PubMed based on the query “miRNA AND fibromyalgia” on the 20 February 2023. Furthermore, we made use of miRDB [44,45]; (target prediction score ≥ 95) to predict gene targets of 37 of the 39 FM-related miRNAs. Targets of hsa-miR-133a and hsa-miR-146a could not be predicted due to lack of specification whether the 3p or 5p miRNA was investigated in the original publication. Ultimately, we checked for overlaps between the predicted gene targets and genes harboring the *p* < 1 × 10^−3^ SNPs. Lists of miRNAs identified from the literature (see Appendix A), their predicted targets (see Appendix A) as well predicted SNP-induced gain or loss of miRNA/target pairs (see Appendix A) are available in the Appendix A.

## 3. Results

### 3.1. Genome-Wide DNA-Polymorphism Associations

The test for association between 48,329 miRNA-related SNPs and fibromyalgia revealed 29 SNPs with *p*-values below our significance threshold of *p* < 1 × 10^−3^, with rs758459 displaying the smallest *p*-value of 0.0001 (see Figure 1, Table 2).

### 3.2. Predicted miRNA/Target Pair Gain or Loss

Mining of the miRNASNP-v3 database revealed that 16 out of the 29 SNPs were predicted to alter binding of altogether 171 unique miRNAs (see Appendix A). Comparison with miRNAs identified from the FM literature exhibited an overlap with hsa-miR-1-3p (MI0000437) whose binding was predicted to be affected by rs2295963 at the *Phospholipid Phosphatase 6* gene (*PLPP6*). Precisely, presence of rs2295963 predicted loss of the hsa-miR-1-3p/*PLPP6* pair. Additionally, rs2295963 was predicted to induce loss of 7 and gain of 4 additional miRNA/*PLPP6* pairs (see Table 3, Figure 2).

### 3.3. Overlap of Genes and Predicted miRNA Targets

A number of 1245 unique genes were predicted as functional targets from 37 literature-extracted miRNAs by miRDB (see Appendix A). Comparison with genes of *p* < 1 × 10^−3^ SNPs revealed an overlap with the *Neurogenin 1* gene (*NEUROG1*), which was predicted to be targeted by hsa-miR-130a-3p and contained the SNP with the lowest *p*-value (rs758459; see Table 2).

## 4. Discussion

FM is a complex chronic pain syndrome characterized by individually varying clinical characteristics and a substantial genetic component [46]. Former GWAS of chronic pain and FM identified multiple SNPs residing in genes that affect central nervous system functioning [47,48] and calcium regulation [47,48,49]. Moreover, an increasing number of miRNAs has been associated with FM pathogenesis (for an overview, see [50]). However, there is still lack of evidence regarding SNPs altering miRNA production and binding in FM. Although SNPs in pre-miRNA loci yield the potential to disrupt production of mature miRNAs, and SNPs in miRNA target genes may alter binding of miRNAs (reviewed by [51]), none have been focused on in FM research.

We have therefore made use of the Axiom miRNA Target Site Genotyping Array to investigate miRNA-related SNPs on a genome-wide level in 170 FM patients and 162 HC. More than 80% of the SNPs that can be genotyped using this array are not found on other genotyping arrays. Accordingly, utilizing this miRNA SNP array was particularly interesting for our research question. To compensate the reduced statistical power that results from our small sample size, we conducted in silico analyses and evaluated our results based on the literature on miRNA-expression in FM. According to our hypothesis, SNPs with *p*-values < 1 × 10^−3^ should possibly have pleiotropic effects (e.g., by affecting miRNA binding sites), influencing molecular signaling pathways and protein expression. In fact, one of the most promising SNPs (rs2295963) identified by our GWAS was predicted to alter binding of hsa-miR-1-3p, a miRNA known to be differentially expressed in FM patients [27]. Moreover, the in silico analysis proposed 11 more miRNAs whose binding may be altered due to rs2295963, suggesting presence of pleiotropic effects. Larger case control studies could replicate this association and give more precise estimates of the effect sizes. Since rs2295963 is located in the 3’UTR for *phospholipid phosphatase 6* (*PLPP6*), a gene that has not yet been described in connection with FM, it would be the logical next step to investigate the effects of this SNP on mRNA stability and PLPP6 protein expression. Luciferase assays and addition of miRNA mimics and antagomirs to cell cultures could provide further insights into the possibilities of the regulatory influence of this SNP. In a further step, it would be interesting to explore which pathways show involvement of PLPP6 and to what extent and under which conditions these pathways contribute to the pathogenesis of FM. In addition, studies of free circulating miRNAs during different phases of pain intensity could provide new insights on how specific miRNAs behave in acute and chronic pain as it was already shown for cell-free DNA under stressful life conditions [52]. These studies could be complemented by drug administration to better understand pharmacological influences on the release of miRNAs in FM pathogenesis.

Further comparisons of predicted miRNA targets and genes of SNPs passing the *p*-value < 1 × 10^−3^ threshold detected an overlap with *NEUROG1* harboring rs758459. Even if *NEUROG1* initially appears to be a promising candidate here, since it assumes important functions as a basic helix-loop-helix (bHLH) transcription factor in the context of neurogenesis, a miRNA-mediated function of rs758459 is probably ruled out here due to its localization in the promoter region of *NEUROG1*. However, a function in the context of *NEUROG1* promoter activity cannot be ruled out and should therefore be examined more closely, e.g., by promoter assays as described elsewhere [53].

Nonetheless, our results are clearly limited by statistical power as well as the number of assessed SNPs. The total sample size of 332 subjects did not allow for discovery of genome-wide significant polymorphisms. Moreover, odds ratios of the *p* < 1 × 10^−3^ SNPs were high due to the small sample size. Additionally, the 48,329 SNPs investigated in this study represent only a fraction of all miRNA-related SNPs [41]. Future studies should therefore not only consider increasing the sample size but also the number of polymorphisms possibly altering miRNA binding and production, allowing for a comprehensive exploration of the miRNA-related genetic variation associated with FM. SNPs in coding regions of FM-related miRNAs are of special interest here, as they may provide an explanation for the already noted aberrant expression patterns of those miRNAs. Moreover, it is important to note that expression of miRNAs is dynamic and tissue-specific [54]. Genome-wide expression studies have shown that miRNAs are expressed differently in postmortem brain tissue depending on psychiatric phenotypes. In vitro studies confirmed that these miRNAs can modulate the protein expression of their specific target genes [55,56]. This is accomplished by epigenetic signatures and can be modulated by external factors [57,58]. Accordingly, results of miRNA-based studies could be further integrated into multi-omics studies to gain a better understanding of pathological altered gene regulation, starting from transcriptional regulation to protein expression [59,60,61,62].

## 5. Conclusions

To conclude, we conducted a GWAS of miRNA-related SNPs in a small FM cohort and identified two novel SNPs based on the respective miRNA expression literature. While rs2295963 was predicted to alter binding of hsa-miR-1-3p as well as binding of 11 additional miRNAs, rs758459 is located within a gene that is targeted by hsa-miR-130a-3p. Even though findings are preliminary, they reveal two promising polymorphisms for functional follow-up studies and a scalable, literature-driven approach to identify further miRNA-related SNPs that may contribute to the complex genetic architecture of FM.

## Figures and Tables

**Figure 1 genes-14-01312-f001:**
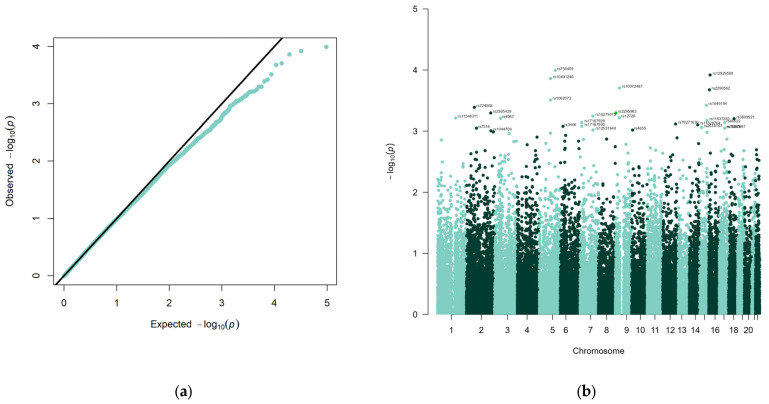
Results of miRNA FM genome-wide association analysis. (**a**) QQ-Plot of genome-wide association analysis; (**b**) Manhattan plot of 48,329 SNPs and their association with fibromyalgia for 170 patients and 162 controls. SNP identifiers from dbSNP 142 are presented for the 29 markers with a *p*-value smaller than 1 × 10^−3^.

**Figure 2 genes-14-01312-f002:**
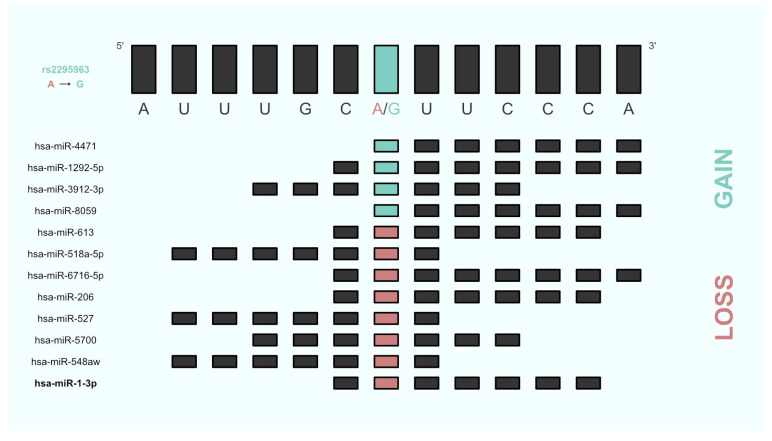
Base pair resolution of rs2295963 induced miRNA/*PLPP6* pair gain or loss. Binding start and end of each miRNA is depicted and based on miRmap v1.1.

**Table 1 genes-14-01312-t001:** Demographic, psychometric, and clinical characteristics.

		FM			HC	
	*M*	*SD*	Range	*M*	*SD*	Range
Age (years)	50.20	9.45	23–68	47.46	15.35	19–81
Pain duration in years	16.99	12.82	0.9–52.6			
CES-D	23.61	7.00	8–44	14.51	4.95	4–32
FSQ						
Symptom Severity Score	9.61	1.95	2–12			
Widespread Pain Index	11.49	4.21	1–19			
FIQ						
Physical functioning	1.46	0.50	0–2.5			
Total	56.69	15.37	23.70–131.85			
CpG						
Pain intensity	72.33	12.63	40–100			
Disability score	65.58	18.39	0–100			
Chronic pain grade	3.35	0.77	1–4			
MPI						
Pain severity	4.07	0.98	1–6			
Interference	4.25	1.14	0.5–6			
Life control	3.16	1.24	0.33–6			
Affective distress	3.54	1.30	0.33–6			
Social support	3.39	1.68	0–6			
Punishing responses	1.32	1.50	0–6			
Solicitous responses	3.28	1.65	0–6			
Distracting responses	2.87	1.42	0–6			
Social activities	2.30	0.98	0.25–5.88			
General activity level	7.53	2.47	2.08–13.18			
PRSS						
Catastrophizing	2.50	1.12	0–4.78			
Coping	3.05	0.79	0.13–4.75			

Note. *M*: mean; *SD*: standard deviation; CES-D: Center for Epidemiologic Studies Depression Scale; FSQ: Fibromyalgia Survey Questionnaire; FIQ: Fibromyalgia Impact Questionnaire; CpG: Chronic Pain Grade Scale; MPI: West Haven-Yale Multidimensional Pain Inventory; PRSS: Pain-related Self Statements Scale.

**Table 2 genes-14-01312-t002:** 29 SNPs passing the *p*-value 1 < 1 × 10^−3^ threshold.

dbSNP RS ID	CHR	BP	A1	OR	*p*	Gene	SNP-Gene-Relationship	NCBI Gene ID
rs758459	5	134872704	C	0.3646	0.0001017	*NEUROG1*	upstream	4762
rs12929500	16	10692832	C	0.4219	0.0001205	*EMP2*	upstream	2013
rs10491246	5	94939777	G	2.394	0.0001377	*ARSK*	UTR-3	153642
rs10972487	9	35482149	T	0.5207	0.0001966	*ATP8B5P*	exon	158381
rs2290562	16	4853816	A	0.5148	0.000211	*GLYR1*	UTR-3	84656
rs1062073	5	94799799	C	2.282	0.0003089	*TTC37*	UTR-3	9652
rs1049194	15	80478645	C	0.4079	0.0003774	*FAH*	UTR-3	2184
rs724568	2	67942480	C	1.773	0.0004073	*ETAA1*	downstream	54465
rs2305429	2	208986385	A	4.495	0.0005041	*CRYGD*	UTR-3	1421
rs2295963	9	4664852	G	2.126	0.0005043	*SPATA6L* *	intron	55064
rs10279573	7	110467221	T	0.4421	0.0005692	*IMMP2L*	intron	83943
rs12726	9	35404840	A	0.5394	0.0006009	*UNC13B*	UTR-3	10497
rs11548311	1	155026871	T	2.431	0.0006086	*ADAM15*	synon	8751
rs4067	3	51738256	A	0.4272	0.0006154	*TEX264*	UTR-3	51368
rs3809921	18	46387889	A	0.5587	0.0006264	*CTIF*	UTR-3	9811
rs11637283	15	86316103	T	1.972	0.0006682	*KLHL25*	intron	64410
rs17167929	7	14187669	C	0.4188	0.0007224	*DGKB*	UTR-3	1607
rs46522	17	46988597	C	0.5838	0.0007262	*UBE2Z*	intron	65264
rs76271636	12	109886494	T	0.264	0.0007639	*KCTD10*	UTR-3	83892
rs11628764	14	92527977	C	0.5439	0.0007893	*ATXN3*	UTR-3	4287
rs17167930	7	14187931	C	0.4235	0.0008299	*DGKB*	UTR-3	1607
rs3406	6	22194573	A	2.258	0.0008349	*CASC15*	intron	401237
rs2618157	15	39844315	A	2.827	0.0008722	*THBS1*	upstream	7057
rs1057897	17	47005509	T	0.5888	0.0008938	*UBE2Z*	UTR-3	65264
rs15563	17	47005193	A	0.5888	0.0008938	*UBE2Z*	UTR-3	65264
rs7316	2	85886013	C	3.151	0.000895	*SFTPB*	UTR-3	6439
rs4655	10	7849688	C	0.5598	0.0009588	*ATP5C1*	UTR-3	509
rs12531640	7	110479535	T	0.4337	0.0009596	*IMMP2L*	intron	83943
rs1044708	2	211298180	T	0.5792	0.0009886	*LANCL1-AS1*	intron	102724820

Note. Additional information is received from the Thermo Fisher Axiom miRNA Target Site Genotyping Array annotation or from dbSNP 156. SNPs are sorted by their *p*-value. * According to the Thermo Fisher annotation, rs2295963 resides in *SPATA6L*. However, dbSNP 156 additionally labels rs2295963 as a *PLPP6* 3-Prime UTR Variant.

**Table 3 genes-14-01312-t003:** Predicted alterations in miRNA binding due to rs2295963.

SNP	miRNA	Binding Start (miRmap)	Binding End (miRmap)	Binding Start (TargetScan)	Binding End (TargetScan)	Target
rs2295963	hsa-miR-4471	4664852	4664858	4664852	4664858	Gain
	hsa-miR-1292-5p	4664851	4664858	4664851	4664858	Gain
	hsa-miR-3912-3p	4664849	4664855	4664849	4664854	Gain
	hsa-miR-8059	4664852	4664858	4664852	4664858	Gain
	hsa-miR-613	4664851	4664857	4664851	4664856	Loss
	hsa-miR-518a-5p	4664847	4664853	4664847	4664852	Loss
	hsa-miR-6716-5p	4664851	4664858	4664851	4664858	Loss
	hsa-miR-206	4664851	4664857	4664851	4664856	Loss
	hsa-miR-527	4664847	4664853	4664847	4664852	Loss
	hsa-miR-5700	4664849	4664855	4664849	4664854	Loss
	hsa-miR-548aw	4664847	4664853	4664847	4664852	Loss
	hsa-miR-1-3p	4664851	4664857	4664851	4664856	Loss

## Data Availability

The data presented in this study are available on request. Individual genetic data cannot be made available due to privacy restrictions. The R scripts used are available on the publicly accessible OSF page of the corresponding author.

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
