# Peer review of "MicroRNA-Related Polymorphism and Their Association with Fibromyalgia"

_genes, 2023, doi:10.3390/genes14071312_

Round 1

Reviewer 1 Report

This study on fibromyalgia is important as it examines miRNA polymorphism. However, the obtained p-value falls below the standard cut-off established in the field. Nonetheless, it is worth mentioning that the authors clearly stated the specific cut-off they used and acknowledged the limitations of their research within the field.

Minor comments

Expand the Manhattan plot, as the current version is too small and difficult to read.

Additionally, provide a diagram illustrating the specific site and interaction 

The variant rs2295963 is predicted to affect the binding of hsa-miR-1-3p.

The variant rs758459 is located within a gene that is targeted by hsa-miR-130a-3p.

For the target prediction score in line 177, please include an opening bracket.

In numerical values, such as in line 159, replace 48.329 with 48,329. Similarly, in line 52, rewrite 2.000 as 2,000.

Please ensure consistency in writing PLPP6, as it is sometimes referred to as PPL6. Correct this inconsistency.

In line 204, indicate whether the disruption leads to a gain or loss and clarify the meaning of "disrupt/enable" in this context, as it is currently unclear.

Author Response

Reviewer 1

Open Review

Quality of English Language

( ) I am not qualified to assess the quality of English in this paper
( ) English very difficult to understand/incomprehensible
( ) Extensive editing of English language required
( ) Moderate editing of English language required
( ) Minor editing of English language required
(x) English language fine. No issues detected

Yes

Can be improved

Must be improved

Not applicable

Does the introduction provide sufficient background and include all relevant references?

( )

(x)

( )

( )

Are all the cited references relevant to the research?

(x)

( )

( )

( )

Is the research design appropriate?

( )

(x)

( )

( )

Are the methods adequately described?

( )

(x)

( )

( )

Are the results clearly presented?

( )

(x)

( )

( )

Are the conclusions supported by the results?

( )

(x)

( )

( )

Comments and Suggestions for Authors

This study on fibromyalgia is important as it examines miRNA polymorphism. However, the obtained p-value falls below the standard cut-off established in the field. Nonetheless, it is worth mentioning that the authors clearly stated the specific cut-off they used and acknowledged the limitations of their research within the field.

We thank reviewer 1 for his constructive and fair criticism. We can certainly understand the weaknesses of our pilot study that he mentioned. However, since there is hardly any literature on genetic variance, miRNA and FM at this point in time, it seems reasonable to us to report the encouraging data of our pilot study, even despite the small cohort size.

Minor comments

Expand the Manhattan plot, as the current version is too small and difficult to read.

We can understand this comment very well. However, the graphic was created exactly according to the Journal's specifications and becomes clearer and more recognizable when zoomed in. Nevertheless, we have provided the Journal with another high-resolution graphic.

Additionally, provide a diagram illustrating the specific site and interaction 

We thank the reviewer for suggesting the idea of an illustrative diagram which presents info on the specific site and the respective miRNA interactions. Correspondingly, a figure was added to the manuscript and sent to the Journal in high-resolution.

The variant rs2295963 is predicted to affect the binding of hsa-miR-1-3p.

            Style of writing was adapted to the reviewer’s proposal.

The variant rs758459 is located within a gene that is targeted by hsa-miR-130a-3p.

            Again, style of writing was adapted.

For the target prediction score in line 177, please include an opening bracket.

            Opening bracket was included.

In numerical values, such as in line 159, replace 48.329 with 48,329. Similarly, in line 52, rewrite 2.000 as 2,000.

            Numerical values were corrected.

Please ensure consistency in writing PLPP6, as it is sometimes referred to as PPL6. Correct this inconsistency.

            PLPP6 is now being used consistently.

In line 204, indicate whether the disruption leads to a gain or loss and clarify the meaning of "disrupt/enable" in this context, as it is currently unclear.

We appreciate the reviewer’s suggestion to clarify whether the disruption resulted in gain or loss of miRNA/target pairs. Accordingly, line 204 and the following were rephrased. 

Submission Date

26 May 2023

Date of this review

05 Jun 2023 15:44:09

Reviewer 2 Report

A very novel study by the authors to elucidate the genetic basis to FM. The authors take a unique approach to address the hypothesis presented and support previous miRNA biomarker quantitative studies in order to prove causation to correlation. The manuscript is well written and concise. The appropriate experimental platforms and approach were utilised to investigate the hypothesis and the data is well interrogated, interpreted and presented. The conclusions drawn are logical.

While the authors themselves highlight the limitations as pilot study on a relatively small cohort, the findings will, in my opinion, stimulate further research in this area (and other complex conditions with a genetic basis) and encourage scientific discourse and debate. Hence, I belief this study will be well cited with a broad readership.

Author Response

Reviewer 2

Open Review

Quality of English Language

( ) I am not qualified to assess the quality of English in this paper
( ) English very difficult to understand/incomprehensible
( ) Extensive editing of English language required
( ) Moderate editing of English language required
( ) Minor editing of English language required
(x) English language fine. No issues detected

Yes

Can be improved

Must be improved

Not applicable

Does the introduction provide sufficient background and include all relevant references?

(x)

( )

( )

( )

Are all the cited references relevant to the research?

(x)

( )

( )

( )

Is the research design appropriate?

(x)

( )

( )

( )

Are the methods adequately described?

(x)

( )

( )

( )

Are the results clearly presented?

(x)

( )

( )

( )

Are the conclusions supported by the results?

(x)

( )

( )

( )

Comments and Suggestions for Authors

A very novel study by the authors to elucidate the genetic basis to FM. The authors take a unique approach to address the hypothesis presented and support previous miRNA biomarker quantitative studies in order to prove causation to correlation. The manuscript is well written and concise. The appropriate experimental platforms and approach were utilised to investigate the hypothesis and the data is well interrogated, interpreted and presented. The conclusions drawn are logical.

While the authors themselves highlight the limitations as pilot study on a relatively small cohort, the findings will, in my opinion, stimulate further research in this area (and other complex conditions with a genetic basis) and encourage scientific discourse and debate. Hence, I belief this study will be well cited with a broad readership.

We are very pleased with this report and thank the expert very much for it. He aptly summarizes the strengths and weaknesses of the study, as we also pointed out in our response to expert 1.

Submission Date

26 May 2023

Date of this review

16 Jun 2023 13:05:05